# In Vitro Evaluation of the Antifungal Effect of AgNPs on *Fusarium oxysporum* f. sp. *lycopersici*

**DOI:** 10.3390/nano13071274

**Published:** 2023-04-04

**Authors:** Karla Lizbeth Macías Sánchez, Hiram Deusdedut Rashid González Martínez, Raúl Carrera Cerritos, Juan Carlos Martínez Espinosa

**Affiliations:** Instituto Politécnico Nacional–UPIIG, Av. Mineral de Valenciana No. 200, Fracc. Industrial Puerto Interior, Silao de la Victoria 36275, Guanajuato, Mexico

**Keywords:** silver nanoparticles, *Fusarium oxysporum*, antifungal activity

## Abstract

The application of nanomaterials in the agri-food industry can lead us to the formulation of new sustainable and effective pesticides for the control of fungi such as *Fusarium oxysporum* f. sp. *lycopersici* (Fol). This is a fungal plant pathogen for the tomato plant. In this work, silver nanoparticles (AgNPs) were synthesized by a green methodology from *Geranium* leaf extract as a reducing agent. The poisoned food technique was used to determine the percentage of inhibition of Fol mycelial growth by the action of AgNPs. They were characterized by transmission electron microscopy (TEM, JEOL JEM-2100, Tokyo, Japan) and ultraviolet-visible spectroscopy (UV-VIS, DU 730 Beckman Coulter, Brea, CA, USA). Five different concentrations of AgNPs (10, 20, 40, 75, and 150 mg/L) were evaluated in vitro in order to determine the minimum inhibitory concentration (MIC) as well as the behavior of their antifungal activity in tomato fruit. Nanoparticles with spherical morphology and average diameters of 38.5 ± 18.5 nm were obtained. The maximum percentage of inhibition on the mycelial growth of Fol was 94.6 ± 0.1%, which was obtained using the AgNPs concentration of 150 mg/L and it was determined that the MIC corresponds to 75 mg/L. On the other hand, in a qualitative way, it was possible to observe an external inhibitory effect in the tomato fruit from the concentration of 10 mg/L. Finally, we can conclude that AgNPs are a viable alternative for alternative formulations applied in the agri-food industry as pesticide solutions.

## 1. Introduction

Phytopathogenic fungus generates some of the main phytosanitary problems in the world, impacting agricultural production [1]. One of the most important is the genus *Fusarium*. These fungi are responsible for a variety of diseases in plantations, such as wilting, blight, and rot, among others [2,3]. *F. oxysporum* is one of the two *Fusarium* species which are considered among the top ten plant pathogens; it is responsible for the most important plant diseases, reaching up to more than 100 host plants [4,5,6], which makes it a target species to control across the land.

The fungi that cause vascular wilt in tomatoes (*Solanum lycopersicum*) are called *F. oxysporum* f. sp. *lycopersici* (Fol) [7,8]. They hinder the transport of water and its nutrients through the colonization of the vascular bundle, which leads to the wilting and death of the plant [7].

To combat these types of problems, agrochemicals people have tested different ways to contain them. The chemical route using systemic and non-systemic fungicides [9], but early or after inaccurate use, has presented harmful side effects such as resistance, environmental wear and tear, and risks to human and animal health [3,10]. On the other hand, it has been shown that some microorganisms can act as antagonists against many phytopathogenic agents, and they are classified as biocontrollers [8]. Other control alternatives of resistant cultivars, whenever they are available, are crop rotation, steam sterilization or dry heat, although these are often economically unviable, and solarization, but the use of large amounts of plastic makes this option an unsuitable solution [11].

An alternative to the problems described above is nanotechnology, which is responsible for manipulating matter on a nanometric scale (10^−9^) in order to generate small particles below 100 nm [12,13]. The chemical, physical, and optical properties of nanoparticles make them highly targeted for certain applications. Some research papers have reported that silver nanoparticles (AgNPs) have a peculiar property in terms of their antifungal, antibacterial, and antiviral mechanism of action, such as the case of *Bacillus subtilis*, *Staphylococcus aureus*, *Pseudomonas aeruginosa*, *Escherichia coli*, *Candida albicans*, *Alternaria alternata*, *Sclerotinia sclerotiorum*, *Macrophomina phaseolina*, *Rhizoctonia solani*, *Botrytis ci-nerea*, HIV-1, and H1N1, among others [14,15,16]. Consequently, AgNPs have become a promising nanomaterial for various specific applications [17]. That pathogenic microorganisms are not resistant to certain nanomaterials makes AgNPs a potential pesticide [18,19]. There are studies that indicate that the intrinsic toxicity of AgNPs depends mainly on factors such as size, shape, surface area, surface charge, solubility, the state of agglomeration of these, and the presence of other compounds [20,21].

Nanoparticles are a viable alternative in the management of sustainable agriculture, as well as to increase agri-food productivity, maintain adequate levels of nutrients in agricultural products, and to mainly reduce the use of herbicides, fertilizers, and pesticides that contain chemical compounds harmful to humans and flora in general [12,22]. Recently, sustainable nanoparticle synthesis methodologies have been investigated in order to avoid or reduce the polluting problems that are commonly generated by different products for agricultural use such as pesticides and herbicides [23,24]. The green synthesis of AgNPs has been an excellent alternative, since this methodology is friendly to the environment by using ecological solvents such as water and ethanol, among others [19,24,25] and where the main factors that determine the size, concentration, and stability of the nanoparticles are the temperature and the concentration of silver salt, in addition to the plant extract that acts as a reducing agent in the reaction kinetics during the nanoparticle synthesis process [25,26]. In this research work, we used the extract of the leaves of the *Geranium* plant as a reducing agent. *Geranium* plants are typical ornamental and aromatic plants used in the cosmetic and sanitary industries and other industries. A study on *Geranium* leaf extract showed the presence of flavonoids, phenolic acid, and cynamic acid as the main reducing compounds that induce the formation of AgNPs from AgNO_3_. The presence of monoterpenes and sesquiterpenes that behave as stabilizers due to C–C groups in their chemical structures [27,28] has also shown. Hence, green synthesis with *Geranium* provides an efficient, economical, and safe method to obtain silver nanoparticles.

The different properties that nanoparticles present on biological systems have been reported. For example, Kim [29] verified the antifungal activity of commercial AgNPs where they observed close to 90% inhibition, in vitro, with a concentration of 100 ppm on Fol. Ashajyothi. [23] carried out antifungical efficiency tests on AgNPs; using *Enterococcus faecali* for their biosynthesis, they determined the MIC of ≤16 µg/mL on *F. oxysporum* MTCC 284. On the other hand, Syed [30] also tested the antifungal effect but by using synthesized AgNPs using an extract of *Justicia peploides* and *Withania coagulans*, indicating an MIC of 11.1 µg/mL using disk diffusion assay. All this implies that there is an opportunity to investigate the antifungal effect of synthesized AgNPs using other plant extracts. Therefore, research into more sustainable alternatives to control phytopathogenics is necessary for the agricultural sector.

In this research work, the in vitro antifungal effect of silver nanoparticles (AgNPs), synthesized with *Geranium* extract and characterized by TEM and UV-Vis, was evaluated. The biological tests were performed on the mycelial growth of Fol by means of the poisoned food technique, the determination of MIC, and directly in tomato fruits.

## 2. Materials and Methods

### 2.1. Biologic Material and Chemical Reagents

The spores of Fol used correspond to the wild strain 4287 race 2, similar to those reported by Wrobel K. et al. [31]. The tomatoes fruits (*Solanum lycopersicum*), *Geranium* Leaves (*Geranium*), and vegetable samples were acquired in commercial establishments in the city of León, Guanajuato, México. The AgNO_3_ was purchased from Meyer reagents (CAS 7761-88-8, Química Suastes, S.A. de C.V., México City, Mexico) and for the microbiology assays, potato dextrose agar (PDA) in Bioxon Laboteca (México City, Mexico) was purchased.

### 2.2. Extract Preparation

An extract was prepared using the fresh leaves of *Geranium*. The leaves were washed and mixed with distilled water up to boiling point for 5 min. Finally, the solution was filtered using a vacuum Kitasato filter (Büchner funnel, Busch R5 RA 0010C vacuum pump, Munich, Germany) and refrigerated at 4 °C [32].

### 2.3. Synthesis of Nanoparticles

A previously reported protocol was followed for the synthesis of the AgNPs [31]. It consisted of a mixture of *Geranium* extract and a 15 mM silver nitrate solution at 85 °C. The mixture was stirred at 435 rpm, and the *Geranium* extract was added in a ratio 9:1. Nanoparticles were used without further purification processes.

### 2.4. Nanoparticles’ Characterization

TEM micrographs of the AgNPs were obtained using a JEM-2100 microscope (JEOL, Tokyo, Japan) equipped with a LaB6 source operated at 80 to 200 kV. The size of all nanoparticles in two image fields were measured to construct the histogram. Further calculation of the average and standard deviation considering a Gaussian distribution was performed. The UV-Vis spectrum was recorded with a DU 730 Beckman Coulter spectrophotometer (PerkinElmer Inc., México City, Mex., Mexico). The analytical determination of silver was performed by an accredited external laboratory (Laquimia lab, San José del Cabo, Mexico) of the Federal Commission for the Protection Against Sanitary Risks, COFEPRIS. The technique employed was inductively coupled plasma (ICP-MS, Laquimia, Irapuato, Gto., Mexico) and the equipment consisted of an EPA 6020A mass spectrometer.

### 2.5. Antifungal Effect on Mycelial Growth In Vitro

The antifungal activity of the nanoparticles against *Fusarium oxysporum* f. sp. *lycopersici* was evaluated using the poisoned food technique [33]. Two hundred µL of AgNPs at different concentrations (10, 20, 40, 75, 100, and 150 mg/L) was added to plates with PDA medium (BD Bioxon MR) by the extension technique. Subsequently, these plates were inoculated with agar discs of 6 mm in diameter with micelia of Fol (precultured for 7 days 27 ± 2 °C) using the poisoned technique. The plates were incubated at 7 ± 2 °C for 7 days. Every 24 h, the growth of the mycelium was recorded radially until the end of the incubation time using a digital vernier (Knova 11726). Sterile distilled water was used as a negative control and the fungicide Captan (N-(trichloromethylthio)-4-cyclohexene-1,2-dicarboximide) at 2500 mg/L [34] was used as a positive control. The percentage of inhibition was obtained by the following formula [29]:Inhibition percentage (%) = ((D − d) × 100)/D(1)
where “D” is the diameter of radial growth of the negative control and “d” is the diameter of radial growth when treated with AgNPs. Analysis of variance (ANOVA) was used to determine the most significant differences between each mycelial fungal growth treatment. Tukey’s method of multiple comparisons (Minitab^®^ version 16, State College, PA, USA) was used to identify the differences between the medium of the inhibition percentage data for each treatment. In addition, a 95% confidence interval (α = 0.05) was used.

### 2.6. Determination of the Minimum Inhibitory Concentration

According to the results of the poisoned food technique, the dilutions of AgNPs to be prepared were determined according to the concentration range where the antifungal effect was observed. To obtain the MIC, tubes were prepared with: 2890 µL of commercial PDB medium, 100 µL of 1 × 10^4^ spores/mL of Fol, and 10 µL of the corresponding AgNPs solution [35]. The tubes were incubated at 27 ± 2 °C for 72 h. At 72 h of incubation, growth was recorded by direct observation. Sterile distilled water was used as a negative control and the fungicide Captan (N-(trichloromethylthio)-4-cyclohexene-1,2-dicarboximide) at 2500 mg/L [34] was used as a positive control. All tests, including the controls, were performed in triplicate.

### 2.7. Antifugal Effect on Tomatoes

Fresh and healthy tomatoes were pretreated by immersing them in 70% ethanol for 5 min. Then, in an aseptic environment, each tomato was inoculated with a mixture of 40 µL AgNPs at concentration ranges from 10–150 mg/L and 1.5 × 10^5^ Fol spores. In addition, distilled water was used as a negative control. The tomatoes were incubated at 27 ± 2 °C for 6 days and fungal growth was recorded every 24 h.

## 3. Results

### 3.1. AgNPs’ Synthesis and Characterization

The synthesis of the AgNPs using the *Geranium* extract produced nanoparticles with an average diameter of 38.5 ± 18.5 nm with spherical and irregular shapes (Figure 1a,b). The absorbance spectrum of the different dilutions of AgNPs shows the characteristic band between 430 and 440 nm (Figure 1c). The quantitative analysis revealed a maximum concentration of 1474.6 mg/L from which dilutions were made for the biological assays.

### 3.2. Evaluation of the Antifungal Effect of the AgNPs

Partial inhibition was observed with the fungicide, as there was a noticeable fungal growth effect after the fourth day (Figure 2a(I)). No growth was presented after 7 days of incubation with a dose of 150 mg/L of the AgNPs (Figure 2a(II)). For concentrations of 10, 20, 40, and 75 mg/L, the antifungal effect increased as well as the concentration of AgNPs (Figure 2a(III,VI)). The control showed a minimum growth of 72.68 ± 0.84 mm and a maximum growth of 74.01 ± 1.05 mm (Figure 2a(VII)).

Fungal inhibition percentages were obtained (Figure 2b) using Formula (1). Progressive mycelial growth inhibition was observed with the concentration of 150 mg/L of AgNPs. This concentration was the one that had the greatest inhibitory effect in the 7 days of incubation, reaching 94.6 ± 0.1%. Another behavior shown is that with concentrations of 20, 40, and 75 mg/L of AgNPs, inhibition increases over time. However, with the concentration of 20 mg/L, this phenomenon can be observed until the third day of incubation due to the fact that since the fourth day, the inhibition rate declined. With concentrations of 40 and 75 mg/L, inhibition remains constant from the fourth and sixth days, respectively. Finally, the concentration of 10 mg/L showed the least mycelial inhibition throughout the incubation period. The positive control showed an inhibition rate of 80.37 ± 0.73% on day three, the largest to be presented in the seven days of incubation (Figure 2b). In addition, the ANOVA statistical analysis showed that there was a significant effect on the inhibition rate due to the concentration of AgNPs. Tukey’s comparisons showed evidence that a higher concentration of green-synthesized AgNPs significantly affects the growth of the fungus; this is due to the groupings obtained in the analysis.

Based on the results obtained in the poisoned food technique (Figure 2a), six concentrations were selected in the range of 25–150 mg/mL to perform the dilution test in the broth to obtain the MIC [36,37]. The results show that with 50 mg/L of AgNPs, fungal growth was presented on the third day of incubation. For concentrations higher than this, no visible fungal growth was observed (Figure 3). Growth in the positive control (fungicide) was not observed either. Therefore, 75 mg/L is the smallest dose where it has no growth with 3 days of incubation (Figure 3f).

Qualitatively, the test on the tomatoes (Figure 4b–e) showed a complete inhibition effect in the areas treated with AgNPs at all their concentrations after seven days of incubation. Interestingly, the maximum concentration (150 mg/L) on day six showed a little bit of small fungal growth compared to the inoculated control only with spores (Figure 4a). Meanwhile, the areas of the tomato inoculated only with Fol spores had a prominent white growth in the tomato surface. In comparison, the tomato control sites showed no growth as expected (Figure 4).

## 4. Discussion

The synthesized AgNPs in this work had an average diameter of 38.5 nm (Figure 1b). These turned out to be larger than some commercial AgNPs (7 to 25 nm) [29], even from some synthesized AgNPs using garlic bulb extract (*Allium sativum*), where they obtained average particle diameters of up to 15.4 ± to 7.9 nm [38]. However, the sizes obtained are similar to those synthesized by Okafor using *Cassia auriculata* leaf extract (diameters between 20 and 40 nm) [39].

Specifically, the fact that the AgNPs obtained in this work are 38.5 nm allows them to be less toxic. Several studies have shown that AgNPs with smaller diameters generate greater toxicity in cells due to their permeability towards the intracellular space [21,40,41]. In addition, the smaller the size, the larger the specific surface area, i.e., the greater the number of atoms available to interact with cells in a physical–chemical way, redox reactions or photochemical reactions, and this larger area is also related to a stronger activity of oxidative stress [16,40,42]. The spherical form resulting from most biosynthesized AgNPs in this study (Figure 1a) is the most typical form reported for green synthesis [28]. In fact, in terms of their shape, being mostly spherical makes them safer, since the triangular shape is the most toxic [42]. Because of what has been mentioned, in this work presenting an approximate size of 50 nm, the possible toxic effect in humans could be less [14,29].

The test results on tomatoes showed qualitatively that AgNPs exhibited inhibitory activity at all tested concentrations. No increase in the inhibitory effect was observed as the concentration of the nanoparticles increased. In addition, it was impossible to visualize the growth inside the tomato without destroying it. Then, only fungal surface growth is observed which was negative for all concentrations.

The synthesized AgNPs presented a UV spectrum with high absorbance values between the range of 350 and 470 nm for the AgNPs (Figure 1c), as reported by other authors [18,38], because AgNPs absorb radiation more intensely in that wavelength range by the transition of electrons [25,39,43]. However, the position and shape of the NPs absorption plasmon depends mainly on the particle size, the dielectric medium, and the interference between the same nanoparticles. Purely spherical NPs spectra have been shown in a single band, and particles of different shapes could result in two or more bands depending on their shape [44]. Therefore, the UV spectrum (Figure 1c) of a band coincides with the spherical morphology of the biosynthesized AgNPs in this study.

Several studies have confirmed the antifungal activity of AgNPs, which is consistent with the results in this work. It was observed that, on the seventh day of incubation, a concentration of 40–75 mg/L showed 81.21–85.61% inhibition like that found in higher concentrations in the literature. For example, Ashraf [45] found that 80–100 mg/L has 85–90% inhibition over the same fungus with AgNPs; it also happens with those synthesized by the green route, but with a particle size of 28 nm on average. Kim [29] obtained 94.1% inhibition on the same fungus, with the same incubation time but at a concentration of 100 mg/L of commercial AgNPs that were 7–25 nm in particle size. The dose of 20 mg/L obtained up to 65.36 ± 5% inhibition in this work, a higher value compared to the work of Kim [29] where 24.7% inhibition was reported at the same concentration of AgNPs. Thus, synthesized AgNPs under the conditions proven in this work show an antifungal effect on Fol as efficiently as previous studies.

Furthermore, Ashajyothi [23] also used synthesized AgNPs with a particle size of 9–130 nm and a spherical form. These were similar to those obtained in this work but with a larger variation in size. They reported inhibition of 72.8% to 60 mg/L of AgNPs over *F. oxysporum* MTCC 284 (from an Indian collection), which is consistent with it being a lower percentage inhibition value compared to the one in of this work of 81.21 ± to 8.52% with the dose of 40 mg/L that was used here (Figure 2b). The differences may be due to the way AgNPs are applicated on the plaque, the strain of the fungus, and the size of the AgNPs.

The fungicide used as a positive inhibitory effect control was Captan, (N-(trichloromethylthote)-4-cyclohexene-1,2-dicarboximide), a non-systemic fungicide, i.e., it is not expected to move through plants, but instead interrupts the fungus life cycle directly. Effectivity has been demonstrated for the treatment of tomato fusariosis caused by Fol [34,44,45,46,47]. Barhate [34] reported 81.25% inhibition on Fol, while research on *F. oxysporum* showed that Captan at 2000 ppm decreases mycelial growth by 71.5% using the same poisoned food technique [48]. These values are greater than the ones obtained in this work of 62.78 ± 21.19%, where, considering the standard deviation, it comes to cover the values in the bibliography. However, the average is still below them. This difference can be explained because the special form (f. sp.) and race were different. However, the broth dilution test (Figure 3) confirmed the antifungal power of Captan before this fungus. The activity of this broad-spectrum fungicide is due to its rapid reaction with thiol groups, as it has side chains containing chlorine, carbon, and sulfur which generates a rapid reaction with the fungus [49].

This study used the concentration of 2500 mg/L (8317 M) of Captan as used in the plate and tube tests. A quantitative criterion was not used in the determination of the MIC due to the optical characteristics of the AgNPs, as it makes this difficult to obtain growth values based on optical density. Because of this, it was chosen to display only qualitative data (Figure 3). The MIC value reported in this work was 75 mg/L of AgNPs, which turns out to be a higher concentration compared to previous studies of the antifungal effect of commercial and biologically synthesized silver nanoparticles [29,37,50]. This difference is mainly attributed to the diameter of the nanoparticles that were used in the reported trials.

While the antifungal mechanism of AgNPs has not been fully described because of the difficulty of studying metal–protein interactions in cells, studies on the antifungal effects of fungi from the genus *Fusarium* with AgNPs synthesized with extracts from other plants have proposed hypothetical mechanisms whereby AgNPs enter the cell and destabilize the cell membrane and enzymes, resulting in the death of the cell membrane. In addition, their small size allows them to enter the nucleus and stop DNA replication through silver ion interactions. They have a higher affinity for phosphorus and sulfur than the DNA phosphate groups have [29]. However, recent research in genomics and transcriptomics on *Fusarium soleni* has shown that AgNPs induce apoptosis in fungal cells by interfering with metabolism, as well as the pathway of signal transduction and the processing of genetic information sensing. In fact, after 24 h of exposure to AgNPs, certain genes related to the metabolism of carbohydrates, fatty acids, amino acids, and nucleotides that were not expressed regained their expression. This indicates the high energy level it cost the cell to survive the stress of AgNPs [51].

Although AgNPs could be used with relative safety in the control of various plant pathogens compared to synthetic fungicides [52], the scientific community has not yet been able to generate a settlement for regulation regarding the use of nanoparticles. In fact, the work of Lekamge [21] shows that there are opposite views regarding the effects that AgNPs cause in living organisms and their toxicity. Hence, more research is needed to gain a better understanding of the toxic potential of AgNPs and the risks that could be taken.

Additional in vivo research would be needed to further assess the fungicide potential of these nanoparticles using plants such as *Geranium*, as used in this work. Testing in the entire tomato plant to study how you would metabolize AgNPs is also needed. A similar studies was conducted by Ashraf [45] whereby they evaluated the use of biosynthesized AgNPs using microwave-mediated *Melia azedarach*, on tomato plants infected with *F. oxysporum* f. sp. *Lycopersici*. In this study, they showed that with 120 mg/L in addition to reducing the severity of fusariosis by up to 90%, it improves growth parameters such as root length. In this way, bases have been established to further investigate more environmentally friendly solutions for the control of tomato fusariosis.

## 5. Conclusions

UV-VIS and TEM characterization suggested the formation of AgNPs by using *Geranium* extract, obtaining an average diameter of 38.5 nm and a spherical morphology. The inhibition test showed that at higher concentration of AgNPs, a higher percentage of inhibition occurs against *F. oxysporum* f. sp. *lycopersici* in vitro on PDA plates and tomato fruits. These nanoparticles proved to be highly efficient compared to the other reported biosynthesized AgNPs due to the size presented here. The highest percentage of inhibition was 94.6 ± 0.1% for 150 mg/L after 7 days of incubation in PDA medium. Moreover, 75 mg/L was the minimum concentration to inhibit the growth of the fungus with the technique of dilution in broth using PDB medium after 3 days of incubation.

## Figures and Tables

**Figure 1 nanomaterials-13-01274-f001:**
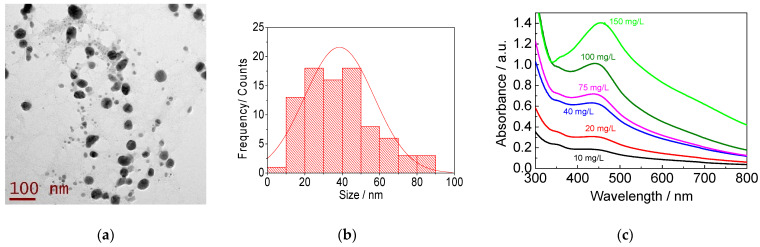
TEM and UV-Vis analysis of the AgNPs: (**a**) micrographs of the AgNPs obtained with the *Geranium* extract observed with a transmission microscope; (**b**) particle size distribution obtained from the micrograph; (**c**) UV-VIS absorption spectrum for the AgNPs at different concentrations.

**Figure 2 nanomaterials-13-01274-f002:**
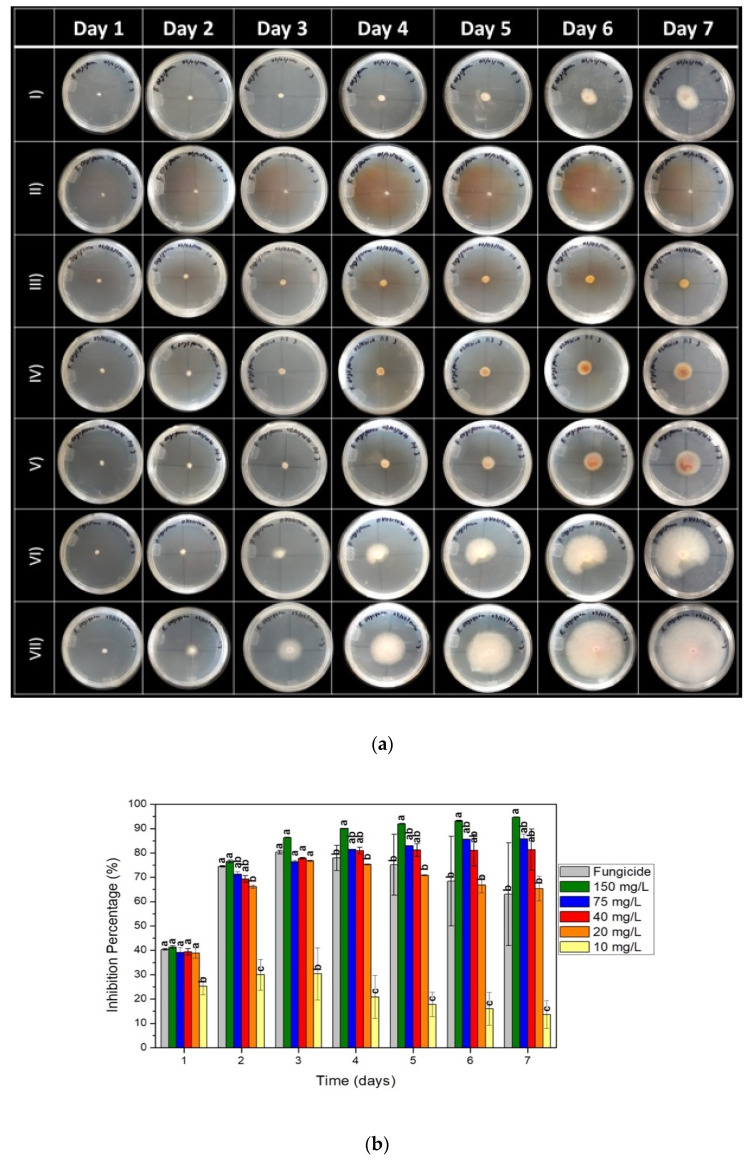
Fol mycelial growth inhibition test treated with AgNPs, synthesized by the green route: (**a**) plates treated with: (**I**) the Fungicide (Captan 2500 mg/L), (**II**) 150 mg/L, (**III**) 75 mg/L, (**IV**) 40 mg/L, (**V**) 20 mg/L, (**VI**) 10 mg/L AgNPs, and (**VII**) sterile distilled water (control); (**b**) percentage of inhibition of mycelial growth over 7 days. Significance testing was performed using analysis of variance ANOVA and Tukey’s test, *p*  <  0.05. Statistically significant differences are indicated by letters (a–c).

**Figure 3 nanomaterials-13-01274-f003:**
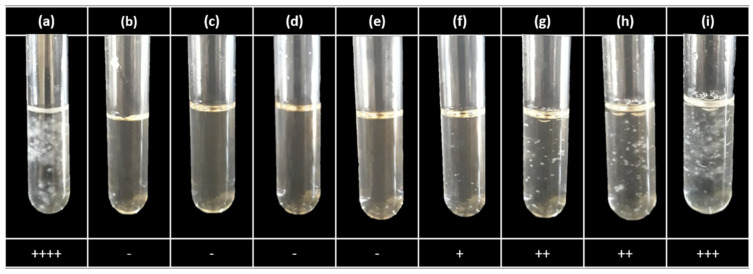
Dilution test in the broth for obtaining CMI from AgNPs, synthesized by the green route, on Fol, after 3 days of incubation. (**a**) Fol; (**b**) fungicide (Captan 2500 mg/L); (**c**) 150 mg/L; (**d**) 100 mg/L; (**e**) 75 mg/L; (**f**) 50 mg/L; (**g**) 40 mg/L; (**h**) 30 mg/L; (**i**) 20 mg/L. ++++ control-like growth (0% inhibition), +++ abundant growth (25% growth inhibition), ++ prominent decrease in growth (inhibition of 50%), + poor growth (inhibition of 75% growth), and (−) no visible fungal growth (100% inhibition) [34].

**Figure 4 nanomaterials-13-01274-f004:**
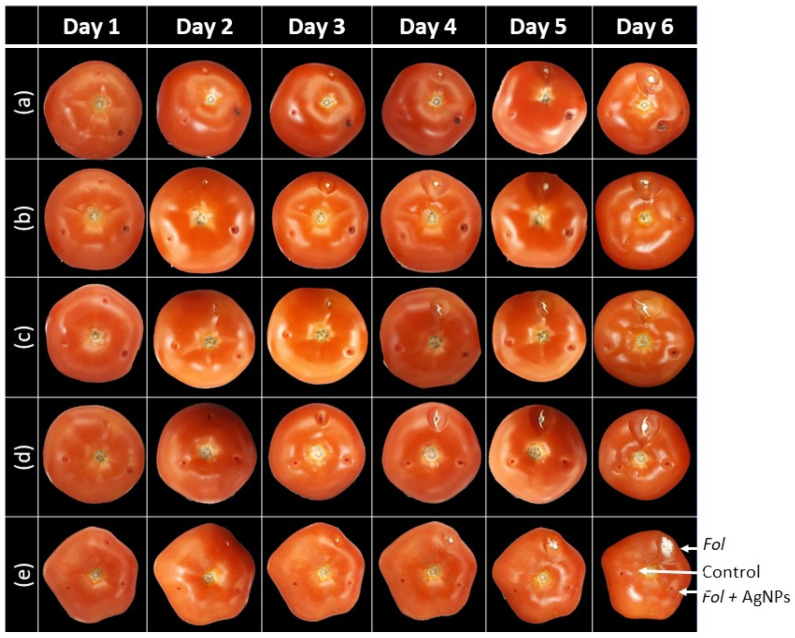
Test of antifungal activity of AgNPs, synthetized by the green route, at different concentrations, in tomatoes. Arrows indicate how each tomato was inoculated: 1.5 × 10^5^ Fol spores (**up**), distilled water as the control (**left**), and 1.5 × 10^5^ Fol spores with AgNPs at a concentration of (**a**) 150 mg/L, (**b**) 75 mg/L, (**c**) 40 mg/L, (**d**) 20 mg/L, and (**e**) 10 mg/L.

## Data Availability

Not applicable.

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
