# Peer review of "In Vitro Evaluation of the Antifungal Effect of AgNPs on Fusarium oxysporum f. sp. lycopersici"

_nanomaterials, 2023, doi:10.3390/nano13071274_

Round 1

Reviewer 1 Report

The Authors presented the study of the effect of the silver nanoparticles on fungus activity in tomato plants. The research aim is interesting, however, the quality of the manuscript is generally poor. First of all, the Authors stated that the Ag NPs were obtained during green synthesis and did not specify it. Then, the AgNPs characterization is poor. Did you use “pure” AgNPs or i.e. surface modified? The strongest side of this manuscript is the studies on tomato plants. Unfortunately in its present form, the paper is not recommended for publishing. Detailed comments are listed below:

The abstract is difficult to read, please rewrite it.

Affiliation - Please add full data, such as full address

Abstract

- how silver can be made from biosynthesized Geranium extract on Fol?

- explain the Position technique

Introduction

This part just brings some general information, and there is no explanation of the major areas presented in this work - what does mean green synthesis, and why AgNPs - pure or modified was used. It is recommended to rewrite the introduction as well. See details below:

- In line 30 what does it mean "reach up to more than 100 hosts"?

- line 51 - nanomaterial - should be nanoparticles

- line 53 - there is no connection with the previous sentence

- line 64 - what do you mean here by green synthesis?

Materials and methods

- separate materials from synthesis, and methods. In its present form, it is difficult to read

Result

- the statement "green synthesis" wasn't proved here. Please specify what you mean.

- how the AgNPs concentration was determined?

- how the AgNPs were purified after synthesis?

Discussion

- line 192 - why did you state that TEM is the most recommended technique for NPs studies? There are various other techniques such as SEM, DLS, etc.. This statement is not necessary and wrong. For details see here:

https://doi.org/10.1021/la301351k

https://doi.org/10.1016/j.foodhyd.2016.09.037

Author Response

Dear reviewer, we attach the response to the comments in a PDF document

Reviewer 2 Report

The introduction is well written. There is quite a lot of information in the introduction regarding fungi of the genus Fusarium. It is obvious that fungi of the genus Fusarium are one of the most significant phytopathogens, which means that the relevance is in order. However, the introduction contains factual errors. For example, the authors write: "However, the traditional way of production of these nanocomposites tends to use toxic chemical reagents that generates pollution by-products, high energy consumption and tend to have low performance [12,21] which limits their extensive application, and paradoxically it would contribute to further increase the problem, so, methods for a more sustainable synthesis have been developed [22,23]." First, the manuscript is not devoted to composites, but to quite specific nanoparticles. Composites consist of two or more materials that do not mix with each other and have a clear distinction. Nanoparticles of any metals obtained using the so-called biological synthesis, as it were, remain nanoparticles and are not composites. The sentence itself is very hard to read. Moreover, the proposal itself is essentially wrong. The method proposed by the authors and similar methods are not used for mass production of nanoparticles. This is just one example, there are many more, I suggest the authors tidy up the introduction. Do not mislead readers! Also, the introduction does not state why geranium extract and silver nanoparticles should be used. With geranium extract, long clarifications are probably unnecessary, you can take any plant. But you need to clarify about silver nanoparticles! For speed, I suggest the authors use the review (10.3390/ph15080968) which details why silver nanoparticles have significant antibacterial and antifungal efficacy. In aqueous solutions, nanosized silver becomes oxides or becomes covered with an oxide film. This also needs to be taken into account! The methods are described quite tolerably. As for the nanoparticles themselves, I personally lack the size distribution. I suggest the author to calculate the size distribution from photographs from an electron microscope. On spectra with plasmon resonance, concentrations must be indicated, not Roman numerals. I recommend making Figure 2a larger, it is very difficult to see something. Very interesting results were obtained on tomato fruits. Can the authors present these data side by side as a histogram? It was very visual! The discussion section of the manuscript is extremely long and needs to be shortened. In general, the manuscript can be recommended for publication after the above problems have been resolved. I am not a native speaker, but it was difficult to read.

Author Response

(The authors gave the same response as above.)

Round 2

Reviewer 1 Report

The Authors strongly improved the manuscript content.

Minor changes – usually editorials are recommended. See details below:

line 64 nanomaterials - should be nanoparticles in my opinion

The synthesis of NPs should be more detailed and presented

Conclusions

- how do UV-Vis spectra might confirm Ag NPs formation? - UV-Vis is a very "weak" technique for confirmation of the structure, elements composition, etc. Here XRD, Raman, or SEM with EDX might be beneficial. Please change the conclusions. Here you can find which technique can be applied where due to nanoparticle characterization.  https://doi.org/10.1016/j.matchemphys.2016.05.03

Author Response

The authors thank the reviewer for the kind comments and suggestions. They helped a lot to improve the manuscript. 

In the attached document we send the updates

Reviewer 2 Report

Can be published as is

Author Response

Thank you for the comments that helped us improve the manuscript.